# Carotid intima-media thickness and risk of atherosclerosis in multiple sclerosis: A cross-sectional study

**Elyar Alizadeh Najmi**[1,2☯], **Zahra Mirzaasgari**[1,2,3,4☯], **Mohammad Reza Motamed**[1,2☯], **Armin Aslani**[ID][5]*

**1** Firouzgar Clinical Research and Development Center (FCRDC), Iran University of Medical Sciences (IUMS), Tehran, Iran, **2** Department of Neurology, Firouzgar Hospital, School of Medicine, Iran University of Medical Sciences, Tehran, Iran, **3** Cellular and Molecular Research Center, Iran University of Medical Sciences, Tehran, Iran, **4** School of Advanced Technologies in Medicine, Iran University of Medical, Tehran, Iran, **5** Department of Community Medicine, Social Determinants of Health Research Center, Faculty of Medicine, Tabriz University of Medical Sciences, Tabriz, Iran

☯ These authors contributed equally to this work.
* Aslard2014@gmail.com

## Abstract

### Background

Multiple sclerosis (MS) is characterized by inflammation and demyelination in the central nervous system. Recent studies have suggested a potential association between MS and an increased risk of atherosclerosis, a systemic vascular disease involving arterial wall thickening. Understanding this relationship is crucial, given the heightened cardiovascular risk observed in MS patients.

### Objective

To investigate factors influencing the development of atherosclerosis in patients with multiple sclerosis (pwMS), focusing on carotid intima-media thickness (CIMT) as a marker.

### Methods

114 pwMS (82 females and 32 males) and 127 control subjects (57 females and 70 males) were included. The mean CIMT between the two groups was compared. Additionally, the effects of annual relapse rate, EDSS (Expanded Disability Status Scale), MS duration, treatment duration, and type of Disease-modifying treatment (DMT) on CIMT were evaluated.

### Results

This study included 241 participants with a mean (SD) age of 41.13 years (10.93). CIMT was significantly higher in pwMS compared to controls ($p < 0.001$). Even after adjusting for age, sex, and BMI (Body Mass Index), CIMT remained significantly higher in the pwMS group ($p < 0.001$). Spearman's correlation analysis revealed significant associations between CIMT and age, BMI, EDSS score, and disease duration (all $p < 0.05$). Additionally, the SPMS (Secondary Progressive Multiple Sclerosis) disease course was significantly associated with

**Data Availability Statement:** All relevant data are within the manuscript and its Supporting Information files

**Funding:** The author(s) received no specific funding for this work.

**Competing interests:** The authors have declared that no competing interests exist.

higher CIMT ($p < 0.001$). Linear regression analysis identified age as the most significant predictor of increased CIMT in pwMS ($p < 0.001$), followed by BMI ($p = 0.054$).

## Conclusion

This study demonstrates a significant association between MS and increased CIMT. Additionally, age emerged as the most prominent predictor of high CIMT in pwMS, followed by BMI. These findings suggest a potential link between MS and increased cardiovascular risk. Further research is warranted to explore the underlying mechanisms and investigate the long-term cardiovascular outcomes in this population.

## Introduction

Multiple sclerosis (MS) is a chronic inflammatory disease of the central nervous system, primarily driven by immune-mediated processes [1]. At the same time, affecting an estimated 2.8 million individuals globally [2]. MS mainly manifests as a debilitating condition, with comorbidities being the primary cause of mortality [3].

Recent investigations suggest a potential association between MS and an elevated risk of cardiovascular diseases, notably atherosclerosis [4–6]. Atherosclerosis, a chronic inflammatory process within the walls of large and medium-sized arteries, contributes to various cardiovascular complications such as ischemic heart disease, stroke, and peripheral vascular disease [7, 8]. Emerging research highlights the presence of an auto-inflammatory component in atherosclerosis, characterized by detecting autoantibodies targeting specific molecules like oxidized low-density lipoprotein(ox-LDL). In the context of MS, it is hypothesized that the chronic inflammation associated with the disease may contribute to the development of atherosclerosis [9]. Despite numerous studies exploring the relationship between MS and atherosclerosis, consensus findings remain elusive [4, 10, 11]. This knowledge gap necessitates further investigation to elucidate the potential impact of MS on atherosclerosis development.

This study hypothesizes that pwMS exhibits an increased prevalence of subclinical atherosclerosis compared to control subjects. Subclinical atherosclerosis is the presence of atherosclerotic disease in one or more arterial territories without any overt signs or symptoms [12]. It represents an early stage of the disease process, where plaque buildup in the arteries is still minimal and has not yet caused noticeable symptoms like chest pain, stroke, or heart attack [13]. To address this hypothesis, we will employ carotid intima-media thickness (CIMT) as a non-invasive surrogate marker for subclinical atherosclerosis [11]. CIMT measurement has been established as a reliable screening tool, demonstrating a strong correlation with increased risks of cerebrovascular and cardiovascular events [14–16]. By comparing CIMT values between pwMS and control subjects, this research aims to gain insights into the potential link between MS and the development of atherosclerosis. The findings will contribute to a better understanding of the cardiovascular implications associated with MS and may pave the way for developing preventive strategies.

## Material and methods

### Participants and methods

In the period between March 31, 2023, and October 3, 2023, we conducted a cross-sectional study in Firouzgar hospital, Tehran, Iran.

We recruited 114 pwMS who were free from diabetes, hypertension, and smoking history. Recruitment took place consecutively at the outpatient unit of the Department of Neurology, Firouzgar Hospital, Tehran, Iran, by neurology residents and attending professors. The recruited patients were either new cases or those checking in for biannual MS visits. The diagnosis of MS was confirmed using the McDonald criteria as outlined by Thompson et al. (2018) [17]. The sample size was determined based on parameters from the study by Quintanilla et al. (2014) [18], with an anticipated 10% dropout rate.

A control group comprising 127 participants selected from hospital staff was included. Control participants were informed about the study through the hospital's intranet system, which provided details on how to express interest in participating. Interested individuals were instructed to contact the research team via email.

Once contact was established, potential participants were invited to a face-to-face meeting or phone call to discuss the study details. During this meeting, informed consent was obtained using a standardized, written document that was translated into the local language. All participants provided written informed consent, and the research adhered to the principles of the Declaration of Helsinki and was approved by the national ethics committee. (reference number: **IR.IUMS.FMD.REC.1401.409**).

## Recruitment and data collection procedures

We captured sex, birthdate, history of smoking, alcohol consumption, dyslipidemia, diabetes, hypertension, ischemic heart disease, myocardial infarction, stroke, and any autoimmune disease in all participants through a self-reported questionnaire.

Participants were instructed to fast for at least 8 hours before blood collection to ensure accurate measurements. Blood samples were collected at the hospital laboratory by trained staff who were unaware of the study's specifics, ensuring participant confidentiality. Blood samples were labelled with unique identifiers to protect participant identities. Data was stored on a fortified, password-protected server, accessible only to authorized personnel. Additional security measures included regular backups and physical security measures to safeguard data integrity.

Various health markers, including fasting blood glucose (FBS), 2-hour postprandial glucose (2hpp), HbA1C, total cholesterol (TC), triglyceride (TG), high-density lipoprotein (HDL), low-density lipoprotein (LDL), complete blood count, Partial Thromboplastin Time (PTT), and Prothrombin Time (PT) were measured. These tests were conducted before biannual visits for pwMS and on the same day as the medical history assessment for the control group.

For diagnostic purposes, FBS levels above 126 mg/dL (7 mmol/L) and 2hpp glucose levels above 200 mg/dL (11.1 mmol/L) indicated diabetes. HbA1C levels exceeding 6.5% (48 mmol/mol) were also classified as diabetes [19]. Dyslipidemia was identified by TC levels above 200, TG levels above 150, or LDL levels above 130 [20]. Coagulopathy was considered if PTT was outside the range of 60–70 seconds or PT was outside the range of 10 to 12.5 seconds [21]. Hypertension was defined as systolic blood pressure above 140 mmHg or diastolic blood pressure above 90 mmHg [22], which was measured by a neurologist. Afterwards, all participants were introduced to the second neurologist for a more comprehensive evaluation.

A fourth-year neurology resident, unaware of the study's objectives, systematically collected comprehensive medical histories using the standard Firouzgar hospital history sheet from all participants, including detailed information on chief complaints, present illness, past medical histories, familial histories, drug histories, social histories, and reviews of systems. Additionally, meticulous examinations were conducted on all participants. Demographic information was obtained using a standardized history sheet. Weight measurements were taken using the

ADE M320000 Approved floor scale (Hamburg, Germany), while height measurements were obtained using a meter. Furthermore, disability levels for pwMS were evaluated using the Expanded Disability Status Scale (EDSS) [23]. Data on MS course type, disease duration, treatment duration, type of disease-modifying treatment (DMT), and total number of annual relapses were obtained from medical records and interviews.

To ensure comparability and minimize the influence of other factors related to atherosclerosis risk, we excluded all individuals with diabetes mellitus, hypertension, dyslipidemia, smoking history, coagulopathy, malignancy, infectious diseases, or known cardiovascular disease from both groups. No alcohol consumption was reported during the data gathering stage.

**CIMT measurement.** The remaining 241 participants underwent CIMT evaluation. CIMT was measured using duplex ultrasound (B-mode) with a Sonosite M-Turbo device equipped with an 8-MHz linear probe (FUJI FILM Sonosite, Washington, USA). A single expert attending professor specialized in neuro-radiology, blinded to the subjects' clinical characteristics, oversaw the examination. Subjects were positioned supine with slightly hyperextended necks for optimal carotid artery access. Real-time B-mode images were captured using a multifrequency 8-MHz linear probe. Three clear images per subject were obtained and analyzed using a cineloop frame grabber. Following established protocols, CIMT was measured in plaque-free common carotid arteries (CCA) areas. The thickness of the intima-media layers was determined to be 10 mm proximal to the CCA bifurcation, adhering to recommended guidelines [24]. According to the American Echocardiographic Association, CIMT values exceeding the 75th percentile for age, race, and sex [25] were considered abnormal and indicative of subclinical atherosclerosis.

## Statistical analysis

The study's findings were presented as mean values with standard deviations (SD) unless otherwise stated. Data was analyzed using SPSS Statistics 24.0 software (IBM, Armonk, NY, USA). To compare the level of CIMT across different categories, we employed the independent samples T-test for binary categories and one-way analysis of variance (ANOVA) for categories with more than two levels. The normality of the data was evaluated using the Kolmogorov-Smirnov test and measures of skewness and kurtosis. Despite slight skewness observed in the data, they were considered approximately normal as both skewness and kurtosis values fell from -1 to 1. The Spearman correlation test was utilized to assess relationships between variables. A linear regression model was used to adjust for age, sex, and BMI in analyzing CIMT differences between groups. Binary logistic regression was also employed to account for age, sex, and BMI when evaluating abnormal CIMT as an indicator of subclinical atherosclerosis. Stepwise linear regression was used to model the impact of variables on CIMT in pwMS. Statistical significance was assumed at a *p*-value less than 0.05.

## Results

### Baseline and clinical characteristics

Baseline characteristics of the study population are presented in Table 1. The study included 241 participants, 114 pwMS, and 127 control subjects. In terms of sex distribution, the majority of pwMS were female (71.09%), while control subjects had a more balanced sex distribution with 55.11% male and 44.89% female. The mean (SD) age of pwMS was 39.55 years (11.23), whereas control subjects had a slightly higher mean (SD) age of 42.55 years (10.49). Regarding BMI, pwMS had a mean (SD) BMI of 25.01 kg/m$^2$ (2.80), compared to control subjects who had a higher mean (SD) BMI of 26.44 kg/m$^2$ (4.99) (Table 1).

**Table 1. Baseline characteristics of the study population.**

| Characteristics | | pwMS (n = 114) | Control subjects (n = 127) | Total (n = 241) | Sig. |
|---|---|---|---|---|---|
| **Sex, No. (%)** | Male | 32 (28.07) | 70 (55.11) | 102 (42.3) | <0001 |
| | Female | 82 (71.09) | 57 (44.89) | 139 (57.7) | |
| **Age,** mean (SD), y | | 39.55 (11.23) | 42.55 (10.49) | 41.13 (10.93) | 0.021 |
| **BMI,** mean (SD), kg.m$^2$ | | 25.01 (2.80) | 26.44 (4.99) | 25.76 (4.15) | 0.012 |
| **EDSS,** mean (SD) | | 2.84 (2.07) | | | |
| **Disease duration,** mean (SD), y | | 9.06 (6.62) | | | |
| **Annual relapse,** mean (SD) No. | | 0.24 (0.61) | | | |
| **Treatment duration,** mean (SD), y | | 6.41 (5.43) | | | |
| **Disease course,** No. (%) | RRMS | 80 (70.2) | | | |
| | SPMS | 26(22.81) | | | |
| | PPMS | 6 (5.3) | | | |
| | PRMS | 2 (1.8) | | | |
| **Latest DMT,** No. (%) | Beta Interferon | 11 (9.6) | | | |
| | Natalizumab | 3 (2.6) | | | |
| | Fingolimod | 7 (6.1) | | | |
| | Teriflunomide | 1 (0.9) | | | |
| | DMF | 4 (3.5) | | | |
| | Rituximab | 41(36.00) | | | |
| | Ocrelizumab | 45 (39.5) | | | |
| | None | 2.00 (1.8) | | | |
| **DMT Duration,** mean (SD), y | Beta Interferon | 6.74 (4.44) | | | |
| | Natalizumab | 1.80(0.84) | | | |
| | Fingolimod | 3.71 (2.81) | | | |
| | Teriflunomide | 3.5 (2.12) | | | |
| | DMF | 3.06 (1.90) | | | |
| | Rituximab | 3.5 (1.85) | | | |
| | Ocrelizumab | 1.16(0.85) | | | |
| | Glatiramer acetate | 2.36 (1.52) | | | |
| | Azathioprine | 10 (-) * | | | |

**Abbreviations**; pwMS: Patients with Multiple sclerosis, No: Number, SD: Standard Deviation, BMI: Body Mass Index, EDSS: Expanded Disability Status Scale, RRMS: Relapsing-Remitting Multiple Sclerosis, SPMS: Secondary Progressive Multiple Sclerosis, PRMS: Progressive-Relapsing Multiple Sclerosis, PPMS: Primary-Progressive Multiple Sclerosis, DMF: Dimethyl fumarate

*Only one user

The mean (SD) EDSS among pwMS was 2.84 (2.07), indicating mild to moderate disability levels on average. The mean (SD) disease duration among pwMS was 9.06 years (6.62), with an annual relapse rate of 0.24 (0.61). The average treatment duration for pwMS was 6.41(5.43) years. Regarding disease course, the majority of pwMS had relapsing-remitting MS (RRMS) (70.2%), followed by secondary progressive MS (SPMS) (22.81%), primary progressive MS (PPMS) (5.3%), and primary relapsing MS (PRMS) (1.8%). Various DMTs were used among pwMS, with the most recent and predominant treatments being Ocrelizumab (39.5%) and Rituximab (36.00%). Beta Interferon was used by 9.6% of pwMS, while 1.8% reported no DMT usage. The mean (SD) duration of DMT usage varied among different agents, with Beta Interferon having the most extended duration of 6.74 (4.44) years and Ocrelizumab the shortest with 1.16 (0.85) years (Table 1).

## CIMT differences between pwMS and control subjects

Since the two groups were not properly matched in terms of age, sex, and BMI—factors that could influence atherosclerosis status and CIMT—we conducted an independent t-test before adjusting for confounders and performed a linear regression analysis to account for these variables.

The mean (SD) CIMT was significantly higher in pwMS compared to control subjects, with a mean (SD) of 0.59 (0.15) mm in pwMS and 0.41(0.11) mm in control subjects ($p < 0.0001$) (Table 2).

In the regression analysis, MS status was a significant predictor of CIMT (B = 0.48, SE = 0.06, $p < 0.001$), indicating that pwMS have increased CIMT compared to controls. While sex (B = 0.009, $p = 0.577$) and BMI (B = 0.003, $p = 0.138$) did not significantly contribute to the model, age showed a substantial effect on CIMT (B = 0.006, $p < 0.001$) (Table 3).

Collinearity diagnostics indicated no issues, as the variance inflation factors (VIF) for all predictors were below 10, ranging from 1.042 to 1.122 (S1 Table in S1 File). Additionally, the model's assumptions were met, and residual statistics confirmed that residuals were normally distributed with a mean of zero and a standard deviation of 0.111 (S2 and S3 Tables in S1 File).

## Subclinical atherosclerosis difference between pwMS and controls

Furthermore, the prevalence of abnormal CIMT (subclinical atherosclerosis) was substantially higher among pwMS compared to control subjects. Among male pwMS, 29.41% had abnormal CIMT, while none of the male control subjects and only 5.26% of females had abnormal CIMT ($p < 0.0001$) (Table 2).

The binary logistic regression analysis revealed that MS status was a significant predictor of abnormal CIMT. PwMS were significantly more likely to have abnormal CIMT compared to controls, with an odds ratio (OR) of 40.80 (95% CI [11.01, 151.10], $p < 0.001$). This suggests that pwMS were approximately 40 times more likely to have abnormal CIMT than those without MS. Although the model was primarily used for covariate adjustment, model fit was also assessed. This model explained between 27.1% (Cox and Snell $R^2$) and 41.4% (Nagelkerke $R^2$) of the variance in IMT status. BMI showed a trend toward significance, but did not reach the conventional threshold (OR = 0.896, $p = 0.059$). sex and age did not significantly contribute to predicting CIMT status (Table 4). The Hosmer and Lemeshow test for the logistic regression model showed no significant lack of fit ($\chi^2 = 5.701$, df = 8, $p = 0.681$). This indicates that the model fits the data well, as there is no evidence of a significant difference between observed and predicted values.

To assess the magnitude of the difference between the two groups, Cohen's d, a measure of effect size, was calculated using the formula: $d = (M_1 – M_2) / S_p$, where $M_1$ and $M_2$ are the means of the two groups, and $S_p$ is the pooled standard deviation. The obtained value of Cohen's d using the following measurements: $n_1 = 114$, $M_1 = 0.59$, $SD_1 = 0.15$ for pwMS and

**Table 2. CIMT difference between pwMS and control subjects without covariant adjustment.**

| Variable | | pwMS | Control subjects | Total | Sig. |
|---|---|---|---|---|---|
| **CIMT,** mean (SD), mm | | 0.59 (0.15) | 0.41 (0.11) | 0.49 (0.16) | <0.0001 |
| **Abnormal CIMT,** No (%) | Male | 15 (46.87) | 0 (0) | 15 (14.70) | <0.0001 |
| | Female | 36 (43.90) | 3 (5.26) | 39 (28.06) | |
| | Total | 51 (44.73) | 3 (2.36) | | |

**Abbreviations**; pwMS: Patients with Multiple sclerosis, CIMT: Carotid Intima-Media Thickness

**Table 3. CIMT difference between pwMS and control subjects, adjusted for age, sex, and BMI.**

| Parameter | B | SE | 95% CI | Sig. |
|---|---|---|---|---|
| Constant | 0.48 | 0.06 | (0.362, 0.600) | <0.001 |
| MS Status* | -0.21 | 0.015 | (-0.23, -0.177) | <0.001 |
| sex | 0.009 | 0.015 | (-0.022, 0.040) | 0.577 |
| age | 0.006 | 0.001 | (0.005, 0.007) | <0.001 |
| BMI | 0.003 | 0.002 | (-0.001, 0.006) | 0.138 |

Abbreviations; MS: Multiple Sclerosis, BMI: Body Mass Index, SE: Standard Error, CI: Confidence Interval
*: having MS vs not having MS.

$n_2 = 127$, $M_2 = 0.41$, $SD_2 = 0.11$ for control subjects, was approximately 1.38. According to Cohen's guidelines, this indicates a large effect size, suggesting a notable difference between the groups.

## Effect of MS disease course on CIMT

An ANOVA was conducted to examine the effect of the MS disease course on CIMT. The results revealed a significant impact of MS disease course on CIMT, $F_{(3, 110)} = 7.362$, $p < 0.001$ (Table 5). A Bonferroni post hoc test was performed to explore further the differences in CIMT between different MS disease courses. Comparisons were made between RRMS and other disease courses. Results indicated a significant mean difference in CIMT between RRMS and SPMS groups (Mean Difference = 0.14, SE = 0.03, $p < 0.001$, 95% CI [0.06, 0.22]). However, no significant differences were found in CIMT between RRMS and other disease courses (Table 6; see S4 Table in S1 File for the complete version).

## Correlation of CIMT with sex, age, BMI, EDSS, and disease duration in pwMS

There was a considerable correlation between age, BMI, EDSS, and disease duration ($p < 0.05$); however, no other remarkable correlation was observed between different variables and CIMT. Other variables, such as annual relapse rate and total treatment duration, did not show significant correlations ($\rho = 0.03$, $p = 0.61$ and $\rho = 0.14$, $p = 0.12$, respectively). Regarding the duration of DMTs, none of the DMTs, including Beta Interferon ($\rho = 0.14$, $p = 0.13$), Natalizumab ($\rho = -0.03$, $p = 0.68$), Fingolimod ($\rho = -0.058$, $p = 0.539$), and others, were significantly correlated with CIMT (Table 7).An independent t-test was conducted to compare CIMT between two sex groups in pwMS; there was no significant difference between males and females ($t = 0.35$, $p = 0.51$) (Table 8).

**Table 4. Binary logistic regression analysis of factors associated with abnormal CIMT status (subclinical atherosclerosis).**

| Predictor | B | SE | Wald | Sig. | OR | 95% CI |
|---|---|---|---|---|---|---|
| MS status (1 = pwMS) | 3.709 | 0.668 | 30.815 | 0.000 | 40.795 | (11.014, 151.099) |
| Sex (1 = Male) | -0.194 | 0.403 | 0.232 | 0.630 | 0.824 | (0.374, 1.815) |
| Age | 0.027 | 0.018 | 2.300 | 0.129 | 1.027 | (0.992, 1.064) |
| BMI | -0.109 | 0.058 | 3.555 | 0.059 | 0.896 | (0.800, 1.004) |
| Constant | 2.034 | 1.510 | 1.813 | 0.178 | 7.642 | - |

Abbreviations; MS: Multiple Sclerosis, BMI: Body Mass Index, SE: Standard Error, OR: Odds Ratio, CI: Confidence Interval

**Table 5. Effect of MS course on CIMT.**

| Source of Variation | Sum of Squares | df | Mean Square | F-statistic | Sig. |
|---|---|---|---|---|---|
| Between Groups | 0.424 | 3 | 0.141 | 7.362 | <0.001 |
| Within Groups | 2.113 | 110 | 0.019 | - | - |

**Abbreviations**; df: degrees of freedom

The analysis assessing the effect of DMT types on CIMT in pwMS revealed no significant associations. The overall model was not significant, with F (9, 104) = 0.402 and $p$ = 0.931, indicating that none of the DMT types significantly influenced CIMT (Table 9).

## Determinants of CMIT in multiple sclerosis

A stepwise linear regression model was employed to explore the predictors of CIMT in pwMS, considering disease duration, disease course, and EDSS as potential predictors while adjusting for age and BMI.

In Model 1, the analysis revealed that EDSS and SPMS were the only significant predictors of CIMT (B = 0.017, SE = 0.008, 95% CI [0.000, 0.034], $p$ = 0.04 and B = 0.087, SE = 0.061, 95% CI [0.001,0.172], $p$ = 0.048, respectively), indicating that higher EDSS scores were associated with increased CIMT values additionally having an SPMS disease course could have an effect on CIMT (Table 10).

In Model 2, the significance of EDSS persisted (B = 0.011, SE = 0.005, 95% CI [0.002, 0.035], $p$ = 0.02). BMI also emerged as a significant predictor of CIMT (B = 0.000, SE = 0.002, 95% CI [0.002, 0.021], $p$ = 0.01), suggesting that higher BMI was associated with increased CIMT values (Table 10).

In Model 3, after further adjustments, age became a significant predictor of CIMT (B = 0.007, SE = 0.001, 95% CI [0.005, 0.010], $p$ < 0.001), along with BMI (B = 0.008, SE = 0.004, 95% CI [0.000, 0.016], $p$ = 0.05) (Table 10). However, none of the models' MS-related variables (i.e., disease duration, SPMS, PPMS, PRMS) were significantly associated with CIMT. RRMS was removed from models due to high collinearity (S8 Table)

The stepwise linear regression analysis results highlight age as the primary predictor of CIMT in pwMS. Specifically, as age increased, CIMT tended to be higher. BMI showed a notable association with CIMT, indicating that higher BMI values were linked to increased CIMT. However, it is important to note that while BMI demonstrated a potential influence, its significance was slightly below the conventional threshold for statistical significance. Notably, the fulfilment of the requisite assumptions for linear regression analysis is documented in the (S6-S8 Tables in S1 File), further bolstering the robustness of our findings.

**Table 6. Bonferroni post hoc test[a].**

| Variable | Comparison | Mean Difference (I-J) | SE | Sig. | 95% CI |
|---|---|---|---|---|---|
| CIMT | RRMS vs. SPMS | 0.14* | 0.03 | 0.000 | (0.06, 0.22) |
|  | RRMS vs. PPMS | -0.12 | 0.06 | 0.25 | (-0.28, 0.04) |
|  | RRMS vs. PRMS | -0.05 | 0.10 | 1.00 | (-0.32, 0.22) |

**Abbreviations**; CIMT: Carotid Intima-Media Thickness, RRMS: Relapsing-Remitting Multiple Sclerosis, SPMS: Secondary Progressive Multiple Sclerosis, PRMS: Progressive-Relapsing Multiple Sclerosis, PPMS: Primary-Progressive Multiple Sclerosis, CI: Confidence Interval
[a] Only comparisons involving the reference group (RRMS) are shown for brevity. Refer to S4 Table in S1 File for the complete version.

**Table 7. Correlation of CIMT and clinical variables in pwMS.**

| Variable | | Co-efficient correlation (ρ) | Sig. |
|---|---|---|---|
| Age | | 0.61 | <**0.001** |
| BMI | | 0.32 | <**0.001** |
| EDSS | | 0.35 | <**0.001** |
| Disease duration | | 0.19 | **0.03** |
| Annual relapse | | 0.03 | 0.61 |
| Total treatment duration | | 0.14 | 0.12 |
| DMT duration | Beta Interferon | 0.14 | 0.13 |
| | Natalizumab | -0.03 | 0.68 |
| | Fingolimod | -0.058 | 0.539 |
| | Teriflunomide | 0.009 | 0.927 |
| | DMF | -0.001 | 0.988 |
| | Rituximab | 0.136 | 0.149 |
| | Ocrelizumab | 0.057 | 0.549 |
| | Glatiramer acetate | 0.014 | 0.882 |
| | Azathioprine | -0.037 | 0.694 |

**Abbreviation;** BMI: Body Mass Index, EDSS: Expanded Disability Status Scale, DMT: Disease Modifying treatment, DMF: Dimethyl Fumarate

## Discussion

In this study, we compared baseline characteristics and CIMT between pwMS and control subjects. A total of 241 participants were included, with 114 pwMS and 127 controls. The mean CIMT was significantly higher in pwMS (0.59 mm) compared to control participants (0.41 mm). Before and after adjusting for potential confounders, including age, sex, and BMI, MS status remained a significant predictor of increased CIMT.

Further stepwise linear regression analysis identified age as the primary predictor of CIMT in pwMS, with older age associated with higher CIMT values. BMI also showed a notable, albeit borderline, association with increased CIMT. These findings, supported by the fulfillment of linear regression assumptions, reinforce the conclusion that age and, to a lesser extent, BMI may contribute to the increased cardiovascular risk observed in pwMS.

Autoinflammation and autoimmunity may play contributing roles in the development of atherosclerosis [26]. The same could be seen in MS since the disease is not solely limited to Central nervous system (CNS) but causes a cascade of systemic inflammations [27]. Numerous studies have indicated vascular dysfunction in the pathogenesis of MS. Due to the inflammatory nature of the disease, a predisposition to vascular diseases may be more frequent in pwMS [28–31]. Observing vascular irregularities in pwMS suggests that MS might lead to additional vascular issues, either due to the widespread inflammation it triggers or through unidentified pathological processes. Inflammation in MS could serve as a shared factor contributing to the development of atherosclerosis. Epidemiological research indicates a heightened likelihood of stroke and

**Table 8. Effect of sex on CIMT in pwMS.**

| Variable | Group | N | Mean (SD) | t | Sig. (2-tailed) | Mean Difference (95% CI) |
|---|---|---|---|---|---|---|
| CIMT | Male | 32 | 0.61 (0.16) | 0.654 | 0.514 | -0.02 (-0.04, 0.08) |
| | Female | 82 | 0.59 (0.14) | | | |

**Abbreviation;** CIMT: Carotid Intima Media Thickness, SD: Standard Deviation, CI: Confidence Interval

**Table 9. Effect of DMT type on CIMT.**

| DMT type | F-statistics | df | Sig. |
|---|---|---|---|
| Overall Model | 0.402 | 9, 104 | 0.931 |
| Beta Interferon | 1.979 | 1, 104 | 0.162 |
| Natalizumab | 0.009 | 1, 104 | 0.923 |
| Fingolimod | 0.141 | 1, 104 | 0.708 |
| Teriflunomide | 0.253 | 1, 104 | 0.616 |
| DMF | 0.094 | 1, 104 | 0.759 |
| Rituximab | 0.944 | 1, 104 | 0.334 |
| Ocrelizumab | 0.164 | 1, 104 | 0.686 |
| Glatiramer acetate | 0.011 | 1, 104 | 0.916 |
| Azathioprine | 0.104 | 1, 104 | 0.748 |

Abbreviation; DMT: Disease Modifying treatment, DMF: Dimethyl Fumarate, df: degrees of freedom

cardiovascular events in pwMS [32]. Factors such as endothelial dysfunction increased oxidative stress in both the CNS and the body, elevated platelet activation, and higher plasma homocysteine levels are linked to the increased risk of vascular events observed in pwMS [33, 34].

**Table 10. Stepwise linear regression model for CIMT with disease duration, disease course, and EDSS as possible predictors, adjusted for age and BMI.**

| | Parameter | B | SE | 95% CI | Sig. |
|---|---|---|---|---|---|
| Model 1 | Constant | 0.52 | 0.026 | (0.471,0.574) | 0.00** |
| | Duration of disease | -0.00004 | 0.002 | (-0.005,0.005) | 0.98 |
| | SPMS | 0.087 | 0.043 | (0.001,0.172) | **0.048*** |
| | PPMS | 0.085 | 0.061 | (-0.037, 0.206) | 0.16 |
| | PRMS | 0.032 | 0.099 | (-0.164,0.227) | 0.74 |
| | EDSS | 0.017 | 0.008 | (0.000,0.034) | **0.04*** |
| Model 2 | Constant | .000 | .002 | (0.009,0.475) | 0.04* |
| | Duration of disease | .068 | .043 | (-0.005,0.004) | 0.85 |
| | SPMS | .072 | .060 | (-0.017,0.153) | 0.11 |
| | PPMS | .021 | .097 | (-0.047,0.192) | 0.23 |
| | PRMS | .019 | .008 | (-0.170,0.213) | 0.82 |
| | EDSS | .011 | .005 | (0.002,0.035) | **0.02*** |
| | BMI | .000 | .002 | (0.002,0.021) | **0.01*** |
| Model 3 | Constant | .104 | .106 | (-0.107,0.315) | 0.33 |
| | Duration of disease | -.002 | .002 | (-0.006,0.002) | 0.33 |
| | SPMS | .027 | .038 | (-0.049,0.104) | 0.47 |
| | PPMS | .012 | .054 | (-00.095,0.12) | 0.81 |
| | PRMS | .022 | .085 | (-0.147,0.191) | 0.79 |
| | EDSS | .005 | .008 | (-0.010,0.020) | 0.51 |
| | BMI | .008 | .004 | (0.000,0.016) | 0.05 |
| | Age | .007 | .001 | (0.005,0.010) | **0.000*** |

Abbreviation; SPMS: Secondary Progressive Multiple Sclerosis, PRMS: Progressive-Relapsing Multiple Sclerosis, PPMS: Primary-Progressive Multiple Sclerosis, EDSS: Expanded Disability Status Scale, BMI: Body Mass Index, SE: Standard Error, CI: Confidence Interval.

Note: RRMS was excluded due to multicollinearity.

**: $p < 0.001$:

*: $p < 0.05$

This study offers new insights into the relationship between MS and atherosclerosis. It is the first of its kind to encompass all subtypes of MS when investigating subclinical atherosclerosis, compared to a control group. Our results indicate that pwMS exhibit significantly higher CIMT and a greater prevalence of subclinical atherosclerosis compared to their control subjects. Specifically, the average CIMT was 0.59 mm among pwMS versus 0.41 mm among control subjects, with subclinical atherosclerosis occurring in 44.73% of pwMS and only 2.36% of control subjects. After controlling age, sex, and BMI as significant risk factors for atherosclerosis as covariates, the difference remained significant between the two groups, suggesting that MS may be associated with an increased risk of atherosclerosis.

Also, we investigate the pwMS to see what factors in these patients contribute the most to atherosclerosis. Age, BMI, disease duration, disease course, and EDSS had meaningful effects. It is evident from the post hoc analysis comparing RRMS and SPMS that patients with SPMS had significantly higher CIMT. This could be explained by the fact that those with SPMS have been living with MS for a longer duration compared to other types of MS; this, in turn, could contribute to the higher CIMT this aligns with the fact that there was a positive meaningful correlation between disease duration and CIMT additionally there was a correlation between EDSS and CIMT this finding it aligned by the fact that the higher the EDSS, the higher the systemic and local inflammation is in the body [35] so this could have contributed to the thickening of the carotid wall. We tried to model using all the influential variables on CIMT in pwMS. Interestingly, only age was meaningful, contributing to the model, and BMI was closely following despite not being meaningful.

Our analysis revealed robust statistical significance associated with the variable "age" across different stages of the study. The high correlation coefficient and significant $p$-value ($<0.001$) observed for "age" indicate its pivotal role in explaining CIMT variability among pwMS. This finding aligns with existing literature, highlighting age as a critical determinant of cardiovascular health and CIMT progression [36, 37]. The consistent significance of "age" in our linear regression models underscores its clinical relevance as a predictor of CIMT despite potential confounding variables.

However, amidst these significant findings, it is imperative to address methodological limitations encountered during our analysis. Notably, the variable "BMI" was nearly dismissed because its $p$-value was marginally higher than 0.05. This observation raises concerns regarding the potential consequences of overlooking critical assumptions of regression modeling, such as normality of residuals and multicollinearity. The non-normal distribution of residuals, coupled with multicollinearity issues, may have obscured the genuine relationship between "BMI" and CIMT, leading to an underestimation of its significance. This highlights the importance of thorough model diagnostics and adherence to regression assumptions to ensure the validity and reliability of regression results.

Furthermore, excluding the variable "RRMS" due to multicollinearity underscores the importance of addressing collinearity issues to obtain reliable coefficient estimates. While this decision mitigated the risk of unreliable estimates, it may have inadvertently affected the interpretation of other predictors in the model.

Despite these challenges, our study contributes valuable insights into the relationship between age and CIMT in pwMS. The significant findings associated with "age" emphasize the need for comprehensive cardiovascular risk assessment and management in pwMS, particularly as they age. Future research endeavors should address methodological limitations, such as regression assumptions, to further elucidate the complex interplay between clinical and demographic factors and CIMT progression in MS.

DMTs have shown to impact the atherosclerosis on some levels. Glucocorticoids, most commonly used during relapses, is closely associated with atherosclerosis [38]. Studies on

the relationship between other DMTs and atherosclerosis have produced mixed results. Some DMTs, such as beta interferon and glatiramer acetate, have been associated with increased cardiovascular disease, in contrast natalizumab had a protective effect [39]. In our study, almost all participants with MS were on DMTs, varying in type and duration. Some patients switched between different treatments due to escalation or de-escalation of treatment. However, the analysis of the nine DMTs—Beta Interferon, Natalizumab, Fingolimod, Teriflunomide, Dimethyl Fumarate, Rituximab, Ocrelizumab, Glatiramer Acetate, and Azathioprine—did not reveal any significant effect on CIMT. The complexity of these treatments, combined with varying durations and combinations, may have influenced CIMT, though this was beyond the scope of the current analysis. A recent study by Omerzu et al. (2024) [40] found no significant differences in CIMT between the RRMS and control groups. Despite having almost similar CIMT in our study (0.572±0.131 vs. 0.59± 0.01), the measured CIMT in the control group was far higher than ours (0.571± 0.114 vs 0.41 ± 0.009). This discrepancy could be explained by the different selection criteria for participants in our study; those smoking and having dyslipidemia were excluded; however, in Omerzu et al. study, nearly one-fourth of pwMS and control groups were smokers additionally mean cholesterol (pwMS: 5.3953± 1.07797 and control group: 5.3071± 1.21927) and LDL (pwMS: 3.435± 0.8941 and control group: 3.298± 0.9667) in both groups which indicate borderline hypercholesteremia ($>5.17$ mmol/L) and above optimal LDL ($>2.6$ mmol/L) which all are known risk factors for atherosclerosis [41].

In the study of Omerzu et al., a homogenous population of RRMS patients was used. This might overlook the effect of long-standing subtypes like SPMS and initially severe ones like PPMS and PRMS Additionally, unlike the study of Omerz et al. pwMS in our study had a longer disease duration (9.06 ± 6.62 years versus 7.29 ± 4.99 years). In similar findings to, our age was the most significant predictor of high CIMT among the participants.

Kemp et al. (2022) [6] studied the association of disability with vascular factors in pwMS. In this study, 51 pwMS were recruited and compared to 25 controls; one of the evaluated variables in this study was CIMT, which indicated no significant difference between the two groups, similar to Omerzu et al. smoking individuals were not exclude 40% of pwMS and 60% of controls were either active or passive smokers and participants lipid profile was not evaluated to exclude those with dyslipidemia it is also worth mentioning that control group had a higher mean (SD) age compared to pwMS (49.6 ± 11.74 versus 47.80 ± 10.66) all the mentioned factors could contribute to the increased IMT in control group. Kemp et al. also evaluated the correlation between CIMT and EDSS, and they discovered a substantial correlation ($r = 0.63$; $p < 0.001$), which aligned with our results.

In a study conducted by Yuksel et al. (2019) [11], 35 RRMS patients were compared with 34 control participants with similar demographic variables similar to our findings; there was a significant difference between RRMS patients and control participants regarding the CIMT ($p<0.001$) with age being significantly correlated to CIMT unlike our study other variables regarding the MS was not evaluated. The measured right CIMT in the Yuksel et al. study (0.6 ±0.09mm) was almost identical to ours (0.59±0.01mm).

It is needless to say that CIMT value depends on racial factors as well [42]; in a study done by Farzan et al. (2024) [4] in Tehran, Iran, 100 pwMS were evaluated for their CIMT, the measured mean CIMT was 0.38 ± 0.2 mm being far smaller than what we have calculated this discrepancy could be explained by the demographical and MS-related factors; pwMS in Farzan et al. study were younger than ours (35.95 ± 9.32 vs. 39.55 ± 11.23 years), included patients had lower disease duration (2.5 ± 8.5 vs. 9.06 ± 6.62 years) and lower EDSS (2 versus 2.84 ± 2.07) all of which could have caused the mentioned difference.

## Limitations

**Study Design:** Our one-time snapshot (cross-sectional design) limits our ability to prove cause and effect or track CIMT changes over time. Long-term studies (longitudinal studies) are needed to understand how MS progression, other heart disease risks, and CIMT relate to each other over time.

**Sample:** Most participants had RRMS with fewer from other types. This may limit applying our findings to people with more progressive MS forms. Future studies should include a wider range of pwMS to represent all disease severities and progressions.

**Unmatched control group:** One significant limitation of this study is the lack of matching between the control group and the pwMS for critical demographic factors, specifically sex, age, and BMI. This mismatch may introduce confounding variables that could influence the outcomes related to CIMT and subclinical atherosclerosis. As a result, the observed differences between the MS group and the control group may not solely reflect the effects of MS but could also be influenced by these unaccounted demographic factors. Future studies should consider matching controls on these variables to enhance the validity and reliability of the findings.

**Unmeasured Factors:** We considered age, sex, and BMI, but other unknown factors (confounders) might have influenced the results. Unmentioned medication use, other health problems (comorbidities), and lifestyle habits weren't fully assessed but could have affected the observed relationships.

## Clinical implications

**Early Risk Assessment:** Our findings suggest including CIMT measurements in routine checks for older pwMS or those with longer disease duration to identify potential atherosclerosis risk.

**Targeted Interventions:** Lifestyle changes (diet, exercise, smoking cessation) and medications for specific risk factors (e.g., high blood pressure, high cholesterol) might be needed to manage cardiovascular risk in pwMS.

## Conclusions

In conclusion, our findings demonstrate a significant association between MS and increased CIMT, a marker of subclinical atherosclerosis. pwMS exhibited significantly higher CIMT compared to the control subjects. Furthermore, age emerged as the most significant predictor of high CIMT within the MS population, followed by BMI. These results suggest that MS may be a potential risk factor for atherosclerosis, highlighting the importance of considering cardiovascular health in MS management. Future research endeavors should aim to elucidate the underlying mechanisms linking MS and atherosclerosis and explore the long-term cardiovascular outcomes in this population.

## Supporting information

**S1 File. This file contains S1-S8 Tables, which present detailed data and analyses related to CIMT and the risk of atherosclerosis in pwMS.** Each table explores specific variables and findings that support the main results of the study.
(DOCX)

**S1 Data. This file includes the raw data used to generate the study's findings on CIMT and atherosclerosis risk in pwMS, supporting the analyses and conclusions discussed in the manuscript.**
(XLSX)

## Acknowledgments

We thank the staff of Firouzgar Hospital for their assistance with this study.

## Author Contributions

**Conceptualization:** Elyar Alizadeh Najmi, Mohammad Reza Motamed.

**Data curation:** Elyar Alizadeh Najmi, Armin Aslani.

**Formal analysis:** Armin Aslani.

**Investigation:** Elyar Alizadeh Najmi, Armin Aslani.

**Methodology:** Mohammad Reza Motamed, Armin Aslani.

**Project administration:** Elyar Alizadeh Najmi, Zahra Mirzaasgari, Mohammad Reza Motamed.

**Resources:** Zahra Mirzaasgari, Mohammad Reza Motamed.

**Software:** Armin Aslani.

**Supervision:** Zahra Mirzaasgari, Mohammad Reza Motamed.

**Validation:** Zahra Mirzaasgari, Mohammad Reza Motamed.

**Writing – original draft:** Elyar Alizadeh Najmi, Armin Aslani.

**Writing – review & editing:** Zahra Mirzaasgari, Mohammad Reza Motamed, Armin Aslani.

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
