## [Decision Letter · Decision Letter 0]

4 Sep 2024

PONE-D-24-16221Carotid intima-media thickness and atherosclerosis in multiple sclerosisPLOS ONE

Dear Dr. Aslani,

Thank you for submitting your manuscript to PLOS ONE. After careful consideration, we feel that it has merit but does not fully meet PLOS ONE’s publication criteria as it currently stands. Therefore, we invite you to submit a revised version of the manuscript that addresses the points raised during the review process.

**ACADEMIC EDITOR: **The manuscript should be cross-checked for typographical errors (capital letters etc.)Keywords should be different from the manuscript title in order to increase its visibilityAvoid use of term “healthy controls”. Those persons probably are not completely healthy, they just don’t have a disease in question. You may use the term “controls” or “control subjects”.Authors stated that cases and controls were matched in terms of age? How is it then possible that there is a 3-year gap between their average age?Were cases and controls matched in terms of sex?Authors should provide more information on the measurement of different laboratory parameters. Were the participants prepared and how? Where were the measurements performed? How were the participants informed about these procedures?Results section should be structured with headings for different paragraphs.==============================

We look forward to receiving your revised manuscript.

Kind regards,

Gorica Maric

Academic Editor

PLOS ONE

2. In the online submission form, you indicated that [The dataset will be accessible upon request.]. 

4. Please include a copy of Table 1,2,3,4,5,6,7 which you refer to in your text on page 5 and 6.

Additional Editor Comments (if provided):

Reviewers' comments:

Reviewer's Responses to Questions

**Comments to the Author**

1. Is the manuscript technically sound, and do the data support the conclusions?

Reviewer #1: Yes

2. Has the statistical analysis been performed appropriately and rigorously? 

Reviewer #1: I Don't Know

3. Have the authors made all data underlying the findings in their manuscript fully available?

Reviewer #1: Yes

4. Is the manuscript presented in an intelligible fashion and written in standard English?

Reviewer #1: Yes

5. Review Comments to the Author

Reviewer #1: Dear Authors,

thank you for the important research and the results shown. Some corrections need to be made.

A shortcoming of the study is that the groups were not matched by gender.

In one discussion paragraph, address the impact of DMT on research results and what other studies have shown on the subject.

Abstract

The method states that 114 people with MS and 127 HCs were included in the research. The results state that a total of 271 people were included in the research. How many people are involved in the research? 241 or 271?

Explain the meaning of the abbreviations BMI, EDSS, SPMS.

Introduction

For the second paragraph of the introduction, it is necessary to put references.

Explain the meaning of "subclinical atherosclerosis".

Material and methods

Emphasize that all of the control group were non-smokers.

Results

It is unclear how it is that the average duration of DMT administration is 6.41, and that the longest administration is beta interferon (whose average duration of administration is 5.1 years).

Discussion

In the first paragraph of the discussion, briefly summarize the main results of this research.

Correct the sentence that atherosclerosis is an autoinflammatory and autoimmune disease.

The sentence in which you say that your results "indicating that MS could be a major risk factor for atherosclerosis" should be corrected and you should not express such a strong position on that issue, given that research still does not have such strong evidence for that claim.

Restrictions

It is necessary to state the limitation regarding groups not matched by gender.

Is consent - not applicable or did the research participants sign an informed consent?

6. PLOS authors have the option to publish the peer review history of their article (what does this mean?). If published, this will include your full peer review and any attached files.

Reviewer #1: No

---

## [Author Response · Author response to Decision Letter 0]

30 Sep 2024

Editor comments

Comments to Author

 The manuscript should be cross-checked for typographical errors (capital letters etc.)

Response: The manuscript has been thoroughly reviewed, and typographical errors, including issues with capital letters, have been corrected.

 Keywords should be different from the manuscript title in order to increase its visibility

Response: The keywords have been revised to enhance the manuscript's visibility. "Intima media thickness" has been removed and replaced with "cardiovascular disease," "Imaging" "Common carotid artery" and "Inflammation."

 Avoid use of term “healthy controls”. Those persons probably are not completely healthy, they just don’t have a disease in question. You may use the term “controls” or “control subjects”.

Response: The term “healthy controls” has been replaced with “controls” or “control subjects” throughout the manuscript, in accordance with the suggestion.

 Authors stated that cases and controls were matched in terms of age? How is it then possible that there is a 3-year gap between their average age?

Response: Thank you for highlighting the discrepancy in age matching between the pwMS and control groups. Upon review, we recognized that while the attending clinician considered the 3-year gap insignificant, the statement regarding matching was not statistically justified and has been removed from the manuscript.

In response to this issue, we performed linear regression analyses adjusting for the key covariates—age, sex, and BMI—given their known influence on atherosclerosis. Even after controlling for these factors, the impact of MS on CIMT remained highly significant (p < 0.001), with CIMT consistently higher in pwMS.

Additionally, we conducted binary logistic regression to assess the presence of subclinical atherosclerosis (abnormal CIMT), adjusting for age, sex, and BMI. The difference between groups remained significant after adjustment.

These revised analyses and findings have been included in the results and supplementary sections.

 Were cases and controls matched in terms of sex?

Response: There was no matching regarding the variable “sex”. Please refer to our response to comment 4 regarding the matching of cases and controls in terms of sex, age and BMI.

 Authors should provide more information on the measurement of different laboratory parameters. Were the participants prepared and how? Where were the measurements performed? How were the participants informed about these procedures?

Response: A comprehensive description of the recruitment process, participant notification, and informed consent procedures has been provided in the manuscript. Participants were informed in advance about the laboratory measurements, including instructions to fast for at least 8 hours prior to blood collection. All laboratory measurements, including fasting blood glucose (FBS), total cholesterol, triglycerides, HDL, LDL, and other markers, were conducted at Firouzgar Hospital laboratory by trained staff following standard preparation protocols. These procedures were carried out to ensure consistency and accuracy, and all results were handled in accordance with ethical guidelines and confidentiality measures.

 Results section should be structured with headings for different paragraphs.

Response: Thank you for your valuable feedback regarding the structure of the results section. In response, we have reorganized the results section to include clear headings for different paragraphs. This restructuring enhances the clarity and navigability of the findings presented in the manuscript.

Reviewer: 1

Comments to Author 

 A shortcoming of the study is that the groups were not matched by gender

Response: Thank you so much for your time and your positive feedback. To compensate this short coming we conducted a set of regression analysis to adjust 3 possible confounding variables; age, sex and BMI. CIMT between wo groups was reassessed by linear regression analysis the difference remained significant after adjusting. As for abnormal CIMT, the surrogate marker of subclinical atherosclerosis, we used binary logistic regression in similar fashion to CIMT, the difference between two groups remained significant as well. 

 In one discussion paragraph, address the impact of DMT on research results and what other studies have shown on the subject. 

Response: Thank you for the suggestion. We have addressed the impact of disease-modifying therapies (DMTs) in the discussion section. Specifically, we note that DMTs have been shown to influence the development of atherosclerosis, with glucocorticoids, commonly used during relapses, being closely associated with an increased risk of atherosclerosis. However, studies on the effects of other DMTs on cardiovascular health have produced mixed results. For example, beta interfrone, and glatiramer acetate have been linked to increased cardiovascular disease risk, while natalizumab has demonstrated a protective effect. In our study, almost all participants with MS were receiving DMTs, and many had switched between different types and durations of therapy over the course of their illness, However, the analysis of the nine DMTs—Beta Interferon, Natalizumab, Fingolimod, Teriflunomide, Dimethyl Fumarate (DMF), Rituximab, Ocrelizumab, Glatiramer Acetate, and Azathioprine—did not reveal any significant effect on CIMT. The complexity of these treatments, combined with varying durations and combinations, may have influenced CIMT, though this was beyond the scope of the current analysis.

 The method states that 114 people with MS and 127 HCs were included in the research. The results state that a total of 271 people were included in the research. How many people are involved in the research? 241 or 271? Response: Thank you for pointing this out. The discrepancy was due to a typographical error. The correct number of participants is 241, with 114 people with MS and 127 healthy controls. This has been corrected in the manuscript.

 Explain the meaning of the abbreviations BMI, EDSS, SPMS

Response: Thank you for your query. The abbreviations BMI, EDSS, and SPMS are written in full in the abstract as Body Mass Index, Expanded Disability Status Scale, and Secondary Progressive Multiple Sclerosis, respectively. We have ensured they are also defined upon first use in the manuscript.

 For the second paragraph of the introduction, it is necessary to put references.

Response: Thank you for your feedback. We have carefully reviewed the second paragraph of the introduction and added all the appropriate references to support the statements made. We appreciate your attention to this detail and have ensured that the references are now correctly cited in the revised manuscript.

 Explain the meaning of "subclinical atherosclerosis".

Response: Thank you for your comment. We have added the definition of subclinical atherosclerosis to the introduction and clarified its relationship with CIMT. Subclinical atherosclerosis refers to the presence of atherosclerotic disease in one or more arterial territories without causing overt signs or symptoms. It represents an early stage of atherosclerosis, where plaque buildup has begun but is not yet severe enough to cause noticeable clinical events such as chest pain, stroke, or heart attack. In this study, we hypothesize that individuals with MS exhibit an increased prevalence of subclinical atherosclerosis compared to control subjects. To test this, we use CIMT as a non-invasive surrogate marker for subclinical atherosclerosis.

 Emphasize that all of the control group were non-smokers.

Response: Thank you for your valuable comment. We have emphasized in the methods section that all participants in the control group were non-smokers. To ensure comparability and minimize the influence of other factors related to atherosclerosis risk, we excluded individuals with diabetes mellitus, hypertension, dyslipidemia, a history of smoking, coagulopathy, malignancy, infectious diseases, or known cardiovascular disease from both groups. This approach allows for a clearer understanding of the outcomes related to our study.

 It is unclear how it is that the average duration of DMT administration is 6.41, and that the longest administration is beta interferon (whose average duration of administration is 5.1 years).

Response: Given that patients used 9 different medications over the course of their disease, with varying durations for each, we have corrected prior miscalculations using the following approach:

For example, in the case of Beta interferon:

n: The total number of patients who used Beta interferon =55

di: The duration of Beta interferon use for patients i (where i = 1, 2, ..., n) 

∑_i^n▒ⅆ_i = sum of Beta interferon use among patients

(∑_i^n▒ⅆ_i )/n=370.60/55=6.738

As for the total DMT duration:

n: The total number of MS patients =114

di: The duration of Rituximab use for patients i (where i = 1, 2, ..., n)

dj: The duration of ocrelizumab use for patients j (where j = 1, 2, ..., n)

.

.

.

 dz: The duration of natalizumab use for patients z (where z = 1, 2, ..., n)

∑(∑_i^n▒ⅆ_i +∑_j^n▒ⅆ_j +⋯+∑_z^n▒ⅆ_z ): sum of all durations=736.60

(∑(∑_i^n▒ⅆ_i + ∑_j^n▒dj +⋯+∑_z^n▒ⅆ_z ))/n=736.60/114=6.46

This formula calculates the average duration of beta interferon use by summing the durations for all patients and dividing by the total number of patients. The 6.46 years reflects the overall average across different DMTs, while the 6.73 years refers specifically to beta interferon.

Since mean of duration is being calculate relative to the total number of specific DMT use, as a result the denominator of the mean fraction is different leading to beta interferon having higher average duration compared to the total average of DMT duration 

 In the first paragraph of the discussion, briefly summarize the main results of this research.

Response: We would like to thank the reviewer for their valuable feedback. In response to the request, we have revised the first paragraph of the discussion to include a brief summary of the main findings. We now highlight the comparison of baseline characteristics and CIMT between pwMS and control subjects, the significant difference in mean CIMT between the two groups, and the persistence of MS as a predictor of increased CIMT after adjusting for confounders. Additionally, we summarize the key insights from the stepwise linear regression analysis, noting age as the primary predictor of CIMT and BMI as having a borderline association.

 Correct the sentence that atherosclerosis is an autoinflammatory and autoimmune disease. 

Response: Thank you for your feedback. We acknowledge that the statement regarding atherosclerosis as an autoinflammatory and autoimmune disease was too definitive. While autoinflammation and autoimmunity may contribute to the development of atherosclerosis, the relationship remains complex and not fully established.

We have revised the sentence to better reflect the current understanding in the literature: "Autoinflammation and autoimmunity may play contributing roles in the development of atherosclerosis."

 The sentence in which you say that your results "indicating that MS could be a major risk factor for atherosclerosis" should be corrected and you should not express such a strong position on that issue, given that research still does not have such strong evidence for that claim.

Response: Thank you for pointing out the need for revision. We agree that the original statement was too strong given the current evidence. We have adjusted the wording to ensure it reflects a more measured interpretation of our findings.

The revised sentence now reads: "suggesting that MS may be associated with an increased risk of atherosclerosis."

 It is necessary to state the limitation regarding groups not matched by gender.

Response: A new paragraph has been added to emphasize the limitation regarding the lack of matching between the groups. The revised paragraph now reads:

"Unmatched control group: One significant limitation of this study is that the control group was not matched with the pwMS group for key demographic factors, particularly sex, age, and BMI. This mismatch may introduce potential confounders that could affect the observed outcomes related to CIMT and subclinical atherosclerosis. Therefore, the differences observed between the MS group and the control group may not be solely attributable to MS but could be partially influenced by these demographic factors. Future research should aim to match control groups on these variables to improve the validity and reliability of the findings."

 Is consent - not applicable or did the research participants sign an informed consent?

Response: There was a misunderstanding, as we initially believed that the journal was requesting the actual informed consents of our participants. To clarify, all participants were thoroughly informed about the study, and written informed consent was obtained from each participant. The revised manuscript now states:

"All participants provided written informed consent prior to their inclusion in the study.

---

## [Decision Letter · Decision Letter 1]

5 Nov 2024

Carotid intima-media thickness and risk of atherosclerosis in multiple sclerosis: a cross-sectional study

PONE-D-24-16221R1

Dear Dr. Aslani,

We’re pleased to inform you that your manuscript has been judged scientifically suitable for publication and will be formally accepted for publication once it meets all outstanding technical requirements.

Kind regards,

Gorica Maric

Academic Editor

PLOS ONE

Additional Editor Comments (optional):

Reviewers' comments:

Reviewer's Responses to Questions

**Comments to the Author**

1. If the authors have adequately addressed your comments raised in a previous round of review and you feel that this manuscript is now acceptable for publication, you may indicate that here to bypass the “Comments to the Author” section, enter your conflict of interest statement in the “Confidential to Editor” section, and submit your "Accept" recommendation.

Reviewer #1: All comments have been addressed

2. Is the manuscript technically sound, and do the data support the conclusions?

Reviewer #1: Yes

3. Has the statistical analysis been performed appropriately and rigorously? 

Reviewer #1: Yes

4. Have the authors made all data underlying the findings in their manuscript fully available?

Reviewer #1: Yes

5. Is the manuscript presented in an intelligible fashion and written in standard English?

Reviewer #1: Yes

6. Review Comments to the Author

Reviewer #1: Dear authors,

Thank you very much for the time and effort you put into answering each question and correcting the request in accordance with the request of the Editor and reviewers.

I believe that all the answers are satisfactory and I suggest that the manuscript be published.

7. PLOS authors have the option to publish the peer review history of their article (what does this mean?). If published, this will include your full peer review and any attached files.

Reviewer #1: No

---

## [Editor Report · Acceptance letter]

7 Nov 2024

PONE-D-24-16221R1 

PLOS ONE

Dear Dr. Aslani, 

I'm pleased to inform you that your manuscript has been deemed suitable for publication in PLOS ONE. Congratulations! Your manuscript is now being handed over to our production team.

Kind regards, 

on behalf of

Dr. Gorica Maric 

Academic Editor

PLOS ONE